# Identifying Support Knowledge Representation for Large Language Models

## Abstract

Large language models (LLMs) have demonstrated remarkable capabilities, yet their knowledge representation mechanism remains opaque. This paper interprets the LLM's attention layers as a notion of Support Knowledge (SK), a learned representation that LLMs use to ground their decisions, exhibiting properties of sparsity, causality, and atomicity. We provide a theoretical analysis of the mathematical connection between margin maximization and the identification of the Support Knowledge. Empirical analyses suggest a two-segment inference mechanism of SK across diverse LLM families (e.g., Qwen3, Llama3.1, ChatGLM4) and tasks (binary QA, textual entailment, and long-document summarization), indicating that there exists layer-wise supporting information units. Interventions on the targeted SK-related neural units reveal their information preserving properties: Specifically on BoolQ, attacking the SK-related neural units would reduce the model accuracy from 82.34% to 27.78%, while random attacks would only lead to a reduced accuracy of 80.43%.

## 1 Introduction

Recent Large Language Models (LLMs) have demonstrated remarkable generalization capabilities across diverse language processing tasks (OpenAI, 2025). However, despite their impressive performance, the internal mechanisms that drive these models' decision-making processes remain largely opaque. While visualization techniques (Vig, 2019) and probing analyses have provided glimpses into their operation, precisely identifying which input tokens critically influence model decisions remains an outstanding challenge in interpretability research.

The process of human reasoning often resembles the search for a minimal set of critical facts. When confronted with a complex text, we intuitively isolate the core evidence, or "support," that is indispensable for a conclusion, while filtering out extraneous details (Van Dijk et al., 1983; Gigerenzer & Gaissmaier, 2011; Shah & Oppenheimer, 2008). This raises a fundamental question: do LLMs independently converge on a similar principle of sparse, evidence-based reasoning? This selective process in LLMs is governed by the attention mechanism, a central component of the Transformer architecture. Recent theoretical work suggests that the optimization of attention is analogous to a Support Vector Machine (SVM) (Cortes & Vapnik, 1995) identifying the most critical data points (Tarzanagh et al., 2024a). Building on this insight, we introduce the concept of Support Knowledge (SK): the sparse, causal, and atomic token subsets that function as the effective support for a model's decision within its attention layers.

In this work, we analyze the core attention mechanism in LLMs, revealing the existence of what we term Support Knowledge: a sparse and causally critical subset of tokens that underpins the model's decisions. We find that this phenomenon is consistent with the margin-maximization (MM) principles of Support Vector Machine theory, which suggests that the model's reliance on SK is not accidental but the result of an optimization process analogous to an SVM identifying its support vectors. The causal role of these SK tokens is confirmed through intervention experiments. Ablating the neural units associated with this sparse set (averaging only 16.87% of the input) causes model accuracy to collapse from 82.34% to 27.78%, a stark contrast to the negligible impact of random interventions (80.44%).

Our work makes several contributions to LLM interpretability:

- We establish a gradient-attention based framework for identifying Support Knowledge in LLMs, bridging classical SVM theory with the transformer architectures through the mathematical connections revealed.
- We empirically demonstrate a general two-segment inference mechanism for Support Knowledge across model depths that characterizes how LLMs process information hierarchically.
- We provide causal validation of Support Knowledge importance through intervention experiments, showing that identified tokens have deterministic influence on model outputs.

## 2 RELATED WORK

### 2.1 NEURAL NETWORK INTERPRETABILITY

The challenge of understanding neural network decision-making has been a fundamental concern since the early days of deep learning. A foundational approach addressed the "black box" criticism of neural networks by introducing validity interval analysis, a method to extract symbolic, human-comprehensible if-then rules from their distributed representations (Thrun, 1994). The field has since evolved significantly with the development of sophisticated post-hoc explanation methods. Researchers have proposed gradient-based attribution techniques such as Integrated Gradients, which satisfies key axioms of sensitivity and implementation invariance through path integration from baseline to input (Sundararajan et al., 2017).

Modern interpretability research has embraced unified theoretical frameworks and model-agnostic approaches. The introduction of SHAP provided a game-theoretic foundation that unifies multiple explanation methods under a single mathematical framework based on Shapley values (Lundberg & Lee, 2017). Complementary work on LIME demonstrated how locally linear models can approximate complex classifier behavior around specific predictions (Ribeiro et al., 2016).

### 2.2 LARGE LANGUAGE MODEL INTERPRETABILITY

A seminal work in LLM interpretability formally equates self-attention with a hard-margin SVM, where attention layers learn to separate optimal from non-optimal tokens through linear constraints (Tarzanagh et al., 2024b). In a complementary line of work, a primal–dual framework derives self-attention from a support vector expansion and proposes attention variants with improved efficiency (Nguyen et al., 2024). Empirical investigations have revealed that BERT's attention heads correspond to linguistic syntax and coreference patterns (Clark et al., 2019b), with representations encoding hierarchical linguistic information—surface features in lower layers, syntactic features in middle layers, and semantic features in higher layers (Jawahar et al., 2019).

Recent work has shifted toward mechanistic interpretability, conceptualizing transformers as computational graphs with information flowing through residual streams and attention mechanisms (Elhage et al., 2021). Researchers have identified induction heads as primary mechanisms for in-context learning (Olsson et al., 2022) and developed sparse autoencoders to address neuron polysemanticity (Bricken et al., 2023). Additional techniques have localized factual associations in middle-layer feed-forward modules, enabling targeted knowledge editing (Meng et al., 2023). Orthogonally, reversed-attention analysis studies the backward pass of attention and shows that gradients induce a sparse, interpretable "reversed" attention pattern that highlights influential tokens during backpropagation (Katz & Wolf, 2025).

While these theoretical and gradient-based analyses of attention provide important insight, how their principles manifest in the practical behavior of modern, deep LLMs remains underexplored. Our work bridges this gap by providing empirical evidence that margin-maximization principles manifest as observable, causally critical computational patterns.

### 2.3 SUPPORT KNOWLEDGE

The discovery that pretrained language models implicitly store vast amounts of factual and linguistic knowledge has spurred a broad field of research aimed at identifying where this knowledge resides and how it can be reliably measured. A research shows that BERT contains relational knowledge

competitive with traditional NLP methods and can recall factual knowledge without fine-tuning through fill-in-the-blank prompts (Petroni et al., 2019). A related study demonstrates that feed-forward layers operate as key-value memories, where keys correlate with textual patterns and values induce distributions over the output vocabulary (Geva et al., 2020). Further research introduces knowledge attribution methods to identify specific neurons that express factual knowledge, finding that their activation is positively correlated with the expression of corresponding facts (Dai et al., 2022).

Probing methods have systematically revealed internal knowledge structures. A study validates that lower Transformer layers carry more type-level lexical knowledge, which is often distributed across multiple layers (Vulić et al., 2020). Another work demonstrates that different layers of contextual representations capture distinct types of linguistic information through edge probing Support Knowledge (Liu et al., 2019). More recent work introduces comprehensive benchmarks to evaluate knowledge recall ability from diverse perspectives, including model size, pretraining strategy, and in-context learning settings (Zhao et al., 2024). While prior work locates static knowledge, we focus on the dynamic, causal mechanism by which LLMs select a sparse subset of tokens for reasoning.

Beyond interpretability, influential tokens have also been used directly as optimization signals in preference optimization and RL-based reasoning (Abdin et al., 2024; Wang et al., 2025). Our work is complementary: we focus on mechanistically identifying a sparse, causally validated set of support tokens that can be plugged into such training schemes.

## 3 METHODOLOGY

### 3.1 TRANSFORMER ATTENTION AND THE SVM

Transformer-based LLMs compute token probabilities through stacked self-attention layers. For an input sequence $X = [x_1, x_2, ..., x_T] \in \mathbb{R}^{T \times d}$, each layer transforms the representation through:

$$h^{(l)} = \text{FFN}(\text{MultiHeadAttention}(h^{(l-1)})) \tag{1}$$

The core computation lies in the attention mechanism. For clarity, consider single-head attention:

$$\text{Attention}(X) = \text{softmax}\left(\frac{XWX^T}{\sqrt{d_k}}\right)XV \tag{2}$$

where $Q, K, V \in \mathbb{R}^{d \times d_k}$ are learned projection matrices. For clarity, we analyze single-head attention where the combined weight matrix can be conceptualized as $W = W_Q W_K^T$. The softmax function normalizes each row:

$$\text{softmax}(z)_i = \frac{\exp(z_i)}{\sum_j \exp(z_j)} \tag{3}$$

Trained language models exhibit a significant pattern: the attention weights $A = \text{softmax}(XQK^TX^T/\sqrt{d_k})$ exhibit sparsity. Certain token positions consistently receive high weights across different contexts, while others are systematically suppressed. This observation indicates that not all tokens contribute equally to the model's predictive capabilities.

The sparsity pattern becomes more apparent when examining token selection. For a given query position with embedding $z \in \mathbb{R}^d$, the attention mechanism assigns the highest weight to

$$\tau^*(z) = \arg \max_{t \in \{1,...,T\}} x_t^\top W z. \tag{4}$$

Eq. 4 explicitly shows that attention selects tokens by maximizing a linear score $x_t^\top W z$. In margin-based classifiers such as SVMs, decisions are likewise driven by linear scores $w^\top x$, where $w$ parameterizes a separating hyperplane. At this linear-scoring level, the attention weight matrix $W$ plays a role analogous to the classifier weight vector $w$: both define directions in representation space that determine which inputs are most influential for the prediction at the target position. This linear structure is similar to those in the SVM and serves as the starting point for our later SVM-inspired interpretation of gradients and attention.

## 3.2 GRADIENT-ATTENTION FRAMEWORK FOR SUPPORT KNOWLEDGE IDENTIFICATION

Building on the margin maximization principle in transformer attention, we now develop a framework for identifying Support Knowledge—tokens that serve as support vectors in the attention mechanism.

**Theoretical Motivation:** We make an analogy between the relationship of gradient magnitude $\|\nabla_{x_t} f(X)\|$ and attention weight $\text{softmax}(XWX^T)_{z,t}$ in Transformers and the primal–dual structure of SVMs.

We consider the unbiased SVM formulation. The optimization is expressed as a minimax problem of the Lagrangian $\mathcal{L}(w, \lambda)$:

$$\max_{\lambda \geq 0} \min_{w} \mathcal{L}(w, \lambda) \iff \min_{w} \max_{\lambda \geq 0} \mathcal{L}(w, \lambda) \tag{5}$$

**Remark:** The bias $b$ is set to 0 as attention relies on relative scores, rendering global bias redundant.

Here, the primal optimization $(\min_w)$ is analogous to the attention mechanism: it updates $W$ so as to increase the alignment score $z^\top W x_t$ for influential tokens. In parallel, we relate the dual variables $\lambda_t$ to per-token gradient magnitudes: tokens with larger $\|\nabla_{x_t} f(X)\|$ have a stronger effect on updating $W$ and thus more strongly constrain the decision boundary, similar to samples with larger dual coefficients in max-margin SVMs.

**Definition 1 (Support Knowledge Score)** *For a token $x_t$ in a sequence, the Support Knowledge score combines gradient and attention information. The score for the token at position $t$ is defined as:*

$$\hat{\Gamma}_t(z) = \alpha \cdot \|\nabla_{x_t} f(X)\|_2 + \beta \cdot a_t \max_{\tau \neq t} (h_t - h_\tau)^\top z \tag{6}$$

*where $\|\nabla_{x_t} f(X)\|_2$ is the gradient norm for the input embedding $x_t$. The second term is a max-margin score, in which $h_t$ denotes a token's hidden state from a given layer, $z$ is the hidden state of the target position, which serves as the target direction in the representation space, $a_t$ is the attention weight on token $t$, and $\alpha, \beta$ are weighting parameters.*

The first component, the gradient magnitude $\|\nabla_{x_t} f(X)\|$, can be interpreted as a dual-like coefficient that quantifies how strong token $t$ influences the decision, analogous to Lagrange multipliers in max-margin methods.

This linear combination suggests an analogy between gradient magnitudes and dual coefficients in margin-based methods such as SVMs. Gradients of the loss with respect to token representations highlight inputs to which the loss is locally most sensitive, while non-zero dual coefficients in the SVM dual problem identify training points that most strongly constrain the decision boundary. In this sense, gradient magnitudes provide a dual-like view of which tokens are most influential.

The second component, $a_t \max_{\tau \neq t} (h_t - h_\tau)^\top z$, is an attention-weighted max-margin score that quantifies a token's contribution to the decision margin. Its structure is derived from SVM principles: the term $(h_t - h_\tau)^\top z$ computes the geometric separation between token $t$ and a competitor $\tau$ in a target direction, while the max operator identifies the margin relative to the closest competitor, analogous to a hard-margin classifier. We modulate this geometric margin by the attention weight $a_t$, grounding the score in the model's actual information flow. A token only receives a high score if it is both geometrically separable and receives significant attention.

Our formulation maintains this structure while adapting to the deep architecture of transformer models, where linear decision boundaries become manifolds in the representation space.

**Definition 2 (Support Knowledge Set)** *Given an input sequence $X$ and target position $z$, the Support Knowledge set $S_{X,z}$ consists of tokens with the most significant scores, identified via a dynamic threshold.*

The identification of this sparse set is guided by the same principles that determine support vectors in SVMs. In classical SVMs, the margin constraints $y_i(w^\top x_i + b) \geq 1$ identify support vectors as the points lying precisely on the margin boundary. For these points, any infinitesimal perturbation would alter the decision boundary, making them maximally influential.

In transformer layers, we measure the influence of token $t$ by the gradient norm $\|\nabla_{x_t} f(X)\|$. Considering the training loss $\ell$ applied to the model output, a first-order Taylor approximation along a scalar coordinate $d_t$ that parameterizes the distance from token $t$'s representation to a local decision boundary yields

$$\|\nabla_{x_t} f(X)\| \propto \frac{\partial \ell}{\partial d_t}. \tag{7}$$

This local sensitivity relation motivates an SVM-inspired view: within transformer layers, tokens with large Support Knowledge scores can be viewed as analogous to support vectors, as they are the elements that most strongly influence the model's prediction at the target position. This principle suggests that if a sparse subset of tokens is functionally analogous to support vectors, then attacks on their associated neural units should be disproportionately more disruptive than on any other equivalent set.

## 4 EXPERIMENTS AND RESULTS

### 4.1 EXPERIMENTAL SETUP

We evaluate our Support Knowledge framework on three representative NLP tasks: BoolQ (Clark et al., 2019a) for binary question answering, RTE (Wang et al., 2019) for sentence-pair textual entailment, and CNN/DailyMail (Hermann et al., 2015) for long-document abstractive summarization. These datasets collectively cover classification and generation, short and long contexts, and single-or-sentence-pair reasoning. The analysis is conducted on Qwen3-8B (Team, 2025). The threshold for selecting support tokens is determined dynamically using an elbow method on the combined scores. Given the set of $N$ positive scores sorted in descending order, $\{s'_j\}_{j=1}^N$, we select the top $k^*$ tokens, where $k^*$ is the number of tokens that minimizes the Euclidean distance to the ideal point $(0, 1)$ on a normalized cumulative score plot. This is formulated as:

$$k^* = \underset{k \in \{1,\ldots,N\}}{\arg\min} \left( \left(\frac{k-1}{N-1}\right)^2 + \left(1 - \frac{\sum_{j=1}^k s'_j}{\sum_{j=1}^N s'_j}\right)^2 \right) \tag{8}$$

For causal validation, we ablate the hidden states ($h_i$) of identified SK tokens using mean-replacement. This standard causal intervention (Meng et al., 2023) removes token-specific information while preserving the overall distributional properties of the sequence representation. This intervention consists of replacing $h_i$ with the sequence mean $\bar{h}$:

$$\tilde{h}_i = \bar{h} \quad \text{where} \quad \bar{h} = \frac{1}{T} \sum_{j=1}^T h_j$$

Intervention impact is measured via model's predictive accuracy. The weighting parameters $\alpha$ and $\beta$ in the SK Score were determined empirically for each model family to maximize intervention impact (details in Appendix B). In the following subsections we use BoolQ as a running example to present detailed analyses.

### 4.2 LOCALIZATION OF SUPPORT KNOWLEDGE

This section analyzes the layer-wise distribution of Support Knowledge (SK) tokens to understand their functional role. We first examine a representative model to detail the SK evolution, then extend the analysis across diverse LLMs to reveal a general two-segment inference mechanism as a general principle.

#### 4.2.1 THE EVOLUTIONARY TRAJECTORY OF SUPPORT KNOWLEDGE

Our analysis framework identifies a sparse subset of Support Knowledge tokens, which on average constitute just 16.87% of the total sequence. To understand how the model selects and utilizes this critical set, we first conduct a micro-analysis of a representative sample from the BoolQ dataset (see Appendix C.2 for full input text), where SK tokens at each layer are identified using the elbow

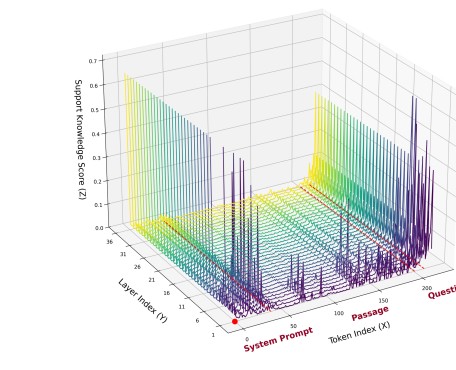

(a) Support Knowledge Score Evolution

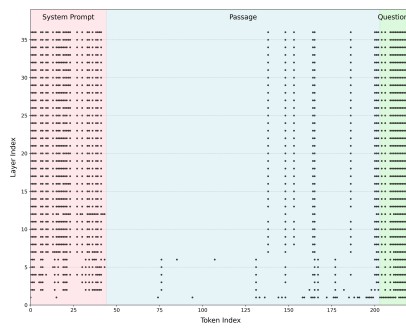

(b) Layer-wise Location of Support Knowledge

Figure 1: Visualization of Support Knowledge (SK) for a representative sample. (a) Shows the evolution of SK Scores (Z-axis) across model layers (Y-axis) and token positions (X-axis). (b) Maps the layer-wise locations of identified SK tokens. These views reveal a two-segment inference mechanism: SKs initially concentrate on Passage content in shallow layers for evidence gathering, before shifting to stabilize on System Prompt elements in deeper layers for task framing.

method defined in Equation 8. The evolutionary trajectory for this sample is visualized in Figure 1, which reveals a two-segment inference mechanism.

Figure 1a reveals that in the earliest layers (1-3), Support Knowledge scores are broadly distributed, indicating an initial exploratory scan. The specific tokens identified, shown in Figure 1b, for instance, at layer 1, the model identifies 46 different SK tokens, including content-specific words from the passage (e.g., "stretches", "Park", "longest"), question-related terms (e.g., "goes", "coast", "interstate"), and system prompt elements (e.g., "assistant"). A significant shift occurs at layers 4-7, where SKs increasingly consist of structural markers. This includes special tokens (`<|im_start|>`) and template words ("Question", "user"). Notably, the answer tokens "true" and "false" first appear as SKs at layer 7, marking the point where the model begins to frame the problem as a binary choice.

By layer 8, the distribution stabilizes into a persistent pattern, with the SK set locking onto a fixed group of tokens as shown in Figure 1b. The SK set settles on a fixed group of tokens, indicating that the model has established a reliable internal representation for the task. Examining these persistent SKs reveals the components of this representation:

- Task-defining elements: "single word", "true", "false", "ONLY"
- Content-specific knowledge: "interstate", "coast", "Highway"
- Structural markers that frame the question-answering process

This case study illustrates a instance of the two-segment inference mechanism: a transition from a broad, content-based set of SK tokens to a focused, task-oriented one. To determine if this structure is a general principle, we then performed an aggregate analysis across a larger set of samples.

Figure 2a presents the average count of SK tokens across layers, categorized by input section. The results from this aggregate analysis confirm the two-segment inference mechanism observed in the individual sample, revealing a general two-segment inference mechanism.

The first part of this process, occurring in the initial layers (approx. 1-7), corresponds to evidence gathering and task construction. In this stage, SK tokens are most abundant in the Passage, but this count rapidly decreases as the number of SKs in the System Prompt rises to a stable, high level.

The second part, which begins around layer 8, is a sustained phase of evidence stabilization. Here, the System Prompt consistently maintains the highest concentration of SKs, while the Passage and Question sections stabilize at lower levels. This stable allocation indicates that after the initial framing, the model operates on a consistent internal representation of the core evidence required for its decision.

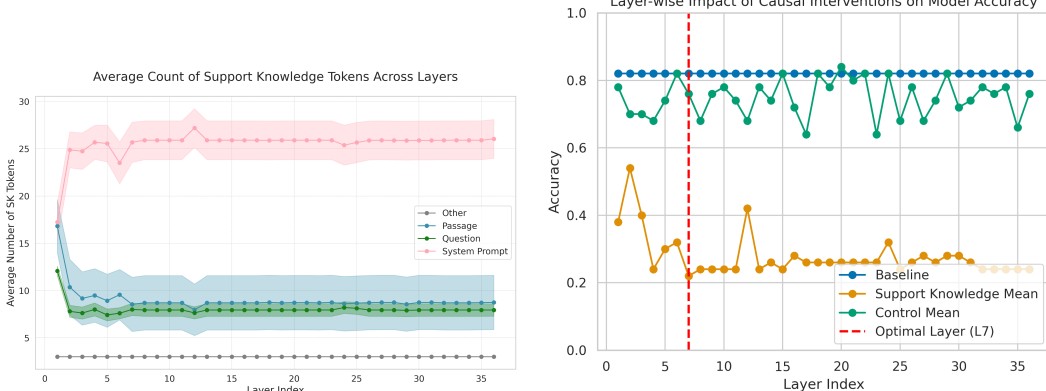

(a) Average count of Support Knowledge across layers for different sections of the input (n=2000).

(b) Layer-wise impact of causal interventions on model accuracy.

Figure 2: Localization and Causal Validation of Support Knowledge (SK). **(a)** The distribution of SK tokens across layers reveals a two-segment inference mechanism, transitioning from evidence gathering in the *Passage* to task framing in the *System Prompt*. **(b)** Mean-replacement interventions confirm the causal importance of SK tokens; targeting SK-associated units (orange) leads to a substantial drop in accuracy, far exceeding the impact of random interventions (green).

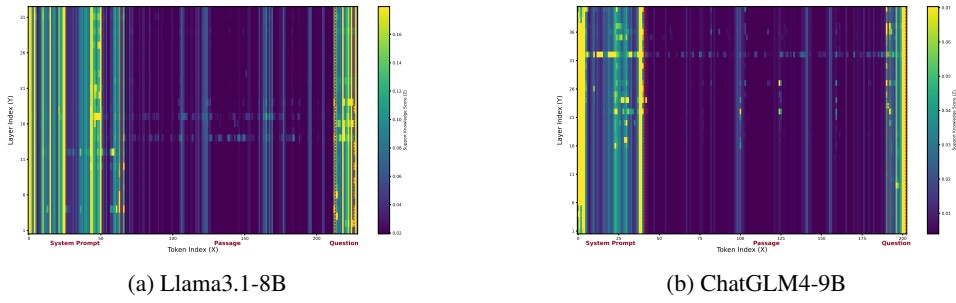

(a) Llama3.1-8B

(b) ChatGLM4-9B

Figure 3: Aggregate Support Knowledge Scores distribution across layers for (a) Llama3.1-8B and (b) ChatGLM4-9B. Both models exhibit a two-segment inference mechanism, but differ in the timing of passage evidence integration.

Quantitatively, adjacent-layer Jaccard overlaps between SK sets reveal a two-segment pattern: in layers 1–7, Adjacent-layer jaccard similarity is low and unstable (mean ≈ 0.52), whereas in layers 8–35 it forms a high plateau (mean ≈ 0.92) with many layers at 1.0. This indicates that SK tokens are reshuffled in the early segment but become effectively fixed in the second segment.

### 4.2.2 CROSS-MODEL AND CROSS-TASK GENERALIZATION OF THE TWO-SEGMENT MECHANISM

To assess the generality of the observed two-segment inference mechanism, we extended our aggregate analysis to Llama3.1-8B (Grattafiori et al., 2024) and ChatGLM4-9B (GLM & et.al, 2024).

Our cross-model analysis, presented in Figure 3, reveals that all tested LLMs adhere to the two-segment inference mechanism previously identified. While the process consistently involves a transition from content-focused exploration to task-focused stabilization, we observe notable variations in its layer-wise timeline.

Llama3.1-8B, for instance, concentrates its focus on passage-based SK tokens in its middle layers (approx. 15-20). In contrast, ChatGLM4-9B postpones this integration of passage-specific evidence to its final layers (approx. 33-40). This divergence suggests that while the two-segment structure

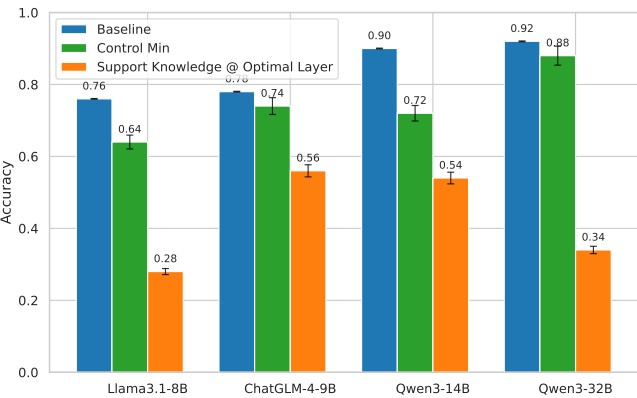

Figure 4: Causal validation of SK across different LLM families. This figure compares baseline accuracy with the impact of two ablation methods: a targeted intervention on SK units and a mean-replacement control. The SK intervention consistently causes a substantial performance collapse, demonstrating its causal necessity. This maximal impact is observed at an optimal layer. Error bars represent the standard deviation of accuracy fluctuation across layers.

of the inference process is a shared principle, its implementation timeline is model-specific, likely reflecting underlying differences in architecture or training objectives. Detailed model-specific analyses are provided in the Appendix D.

Beyond cross-model comparisons on BoolQ, we observe a similar two-segment pattern on RTE and CNN/DailyMail. On RTE, the mean Jaccard overlap between SK sets at adjacent layers is 0.62 across the early layers (approx. 1–7), but rises to 0.97 from layer 8 onward, indicating a transition from rapidly changing to nearly frozen SK sets. On CNN/DailyMail, the overlaps in early layers (approx. 1–7) remain around 0.69–0.77, while in later layers (approx. 8–36) they are close to 1.0 and predominantly above 0.97, reflecting a stable configuration of SK tokens that supports the second inference segment (see Appendix I for full curves).

## 4.3 Intervention Experiments

Drawing a parallel to SVM theory, where attacks on support vectors are most disruptive, we analyze the causal importance of neural units associated with Support Knowledge (SK) tokens. We attack these units at the embedding layer via mean replacement to measure their causal necessity from their point of origin. For SK tokens identified at a given layer $L$, their corresponding embeddings are replaced by the sequence mean. The impact of this targeted attack is quantified by the degradation in model performance, compared against an unmodified baseline and a control group where an equivalent number of random neural units are attacked.

The results, presented in Figure 2b, validate our hypothesis regarding the disproportionate causal importance of SK-associated neural units. Intervening on these units (orange curve) causes a greater reduction in model accuracy than intervening on units associated with random tokens (green curve). This difference is confirmed as statistically significant by a paired t-test ($t(35) = -33.58, p < 0.001$), confirming they constitute irreducible components of the model's inference process. The intervention impact is greatest at Layer 7, which corresponds to the transition point from task construction to evidence evaluation identified in our localization analysis (cf. Figure 2a).

To determine if the observed causal importance of these neural units is a general principle, we extended our analysis to a diverse set of LLMs, including different architectures and scales. As shown in Figure 4, targeted ablation of SK units results in a large reduction in model accuracy. Across all tested LLM families, this performance degradation is consistently more severe than that from the mean-replacement control. The detailed layer-wise analysis for each model is provided in the Appendix D.

## 4.4 BASELINE COMPARISON

To evaluate the effectiveness of gradient and attention signals in identifying Support Knowledge tokens, we compare it with three alternative token identification strategies: a Positional Heuristic (using token position),Pure Attention (using only attention weights), and Pure Gradient (using only gradient magnitudes). Each method's effectiveness is measured by applying mean-replacement intervention to the neural units associated with the SK tokens and observing the resulting accuracy change.

Table 1: Comparative Analysis of Token Selection Strategies via Intervention Impact on BoolQ, RTE, and CNN/DailyMail

| Token Selection Strategy | BoolQ $\Delta$ | RTE $\Delta$ | CNN/DM R-1 | R-2 | R-L |
|---|---|---|---|---|---|
| No Intervention (Baseline) | 0.00% | 0.00% | 21.46 | 4.91 | 19.10 |
| Random Selection | -1.90% | -1.81% | 21.58 | 4.88 | 18.95 |
| Positional Heuristic (Last 18%) | -30.64% | -1.44% | 22.13 | 4.54 | 19.69 |
| Pure Attention | -18.17% | -1.44% | 19.85 | 4.39 | 17.79 |
| Pure Gradient | -45.17% | -40.07% | 15.98 | 2.50 | 14.71 |
| **Gradient-Attention (Ours)** | **-54.56%** | **-40.43%** | **11.04** | **1.94** | **10.45** |

Table 1 presents the post-intervention performance of different token selection strategies across BoolQ, RTE, and CNN/DailyMail. On BoolQ, random interventions have a minimal impact (80.44% vs. 82.34% baseline), and positional heuristics yield a moderate drop to 51.70%, while single-signal methods are more disruptive (64.17% for pure attention and 37.17% for pure gradient). Our combined gradient-attention score causes the most severe collapse to 27.78%, a statistically significant degradation relative to the strongest baseline (Pure Gradient; McNemar's test, $p < 0.01$). On RTE, the same ordering holds: random and positional selection keep accuracy close to the 88.45% baseline, whereas pure gradient and our method reduce it to 48.38% and 48.01%, respectively. On CNN/DailyMail, the ROUGE-1/2/L columns show that random and positional perturbations barely change or even slightly improve summarization quality, and our gradient-attention score yields the most pronounced degradation. Taken together, these results indicate that the SK tokens singled out by our method form the subset of representations that is most causally important for model behavior across both classification and generation tasks.

## 5 CONCLUSION

This work provides empirical evidence that the attention mechanisms of LLMs operate on a margin-maximization principle, analogous to the learning of support vectors in SVMs. We show that Support Knowledge tokens constitute sparse, causally critical units of reasoning: ablating the neural units associated with these tokens leads to a collapse in performance, and the same two-segment inference pattern for SK appears consistently across architectures and across binary QA, textual entailment, and long-document summarization. These results suggest that SK is an invariant computational principle with architecture-dependent realizations in depth and layer-wise timing. We treat Support Knowledge primarily as a conceptual and diagnostic tool for understanding how LLMs reason. Future work may leverage SK to design more targeted editing and distillation procedures, and to investigate how SK-based reasoning manifests in multi-hop and multi-modal settings.

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

## A  ALGORITHM FOR CAUSAL INTERVENTION EXPERIMENTS

The overall procedure for our causal intervention experiments is outlined in Algorithm 1. This algorithm details the main experimental loop, covering the layer-wise process of identifying Support Knowledge (SK) and control tokens, performing interventions, and evaluating the impact on model accuracy. To provide a complete picture of the methodology, we further detail the two core components of this loop in separate algorithms. Algorithm 2 specifies the procedure for SK identification, while Algorithm 3 outlines the mean-replacement intervention mechanism.

## B  MODEL-SPECIFIC WEIGHTING FOR SK SCORE CALCULATION

The Support Knowledge (SK) Score is calculated as a weighted linear combination of a gradient term and an attention-based max-margin term. The parameters $\alpha$ and $\beta$ control the relative contribution of the gradient and attention signals, respectively, and are constrained such that $\alpha + \beta = 1$. As we observed that the optimal balance between these components is not uniform across different model architectures, we determined model-specific weights empirically.

These weights were calibrated with the objective of maximizing the accuracy drop during our mean-replacement intervention at each model's most sensitive layer. This procedure ensures that the selected weights most precisely isolate the set of tokens that are causally critical for that model's reasoning process.

The finalized weights used for all experiments in this paper are presented in Table 2.

Table 2: Model-specific weights for the SK Score components.

| Model | $\alpha$ (Gradient) | $\beta$ (Attention) |
|---|---|---|
| Qwen3-8B | 0.2 | 0.8 |
| Qwen3-14B | 0.3 | 0.7 |
| Qwen3-32B | 0.3 | 0.7 |
| Llama3.1-8B | 0.7 | 0.3 |
| ChatGLM4-9B | 0.8 | 0.2 |

The resulting weights reveal a distinct pattern. The Qwen3 model family consistently exhibits a stronger reliance on the attention component ($\beta > \alpha$). In contrast, Llama3.1-8B shows a clear preference for the gradient signal, while ChatGLM4-9B demonstrates the most pronounced reliance on gradients. This divergence suggests that underlying architectural differences or pre-training objectives may lead models to encode critical information differently, with some prioritizing the geometric separation captured by attention and others prioritizing the optimization constraints revealed by gradients.

---

**Algorithm 1** Layer-wise Causal Intervention Procedure

---

1: **Input:** Model $M$, Tokenizer $T$, Dataset $D_{sub}$, Layers $L$
2: **Output:** Layer-wise accuracies $R$ (Baseline, SK, Control)
3: Initialize results table $R \leftarrow \emptyset$
4: **for** each $layer \in L$ **do**
5:     Initialize counters: $c_{base}, c_{sk}, c_{ctrl} \leftarrow 0, 0, 0$
6:     **for** each $sample \in D_{sub}$ **do**
7:         $prompt \leftarrow$ FormatPrompt($sample$)
8:         $inputs \leftarrow T(prompt)$
9:         **// Step 1: Baseline Evaluation**
10:        $logits_{base} \leftarrow M(inputs)$
11:        $pred_{base} \leftarrow$ GetPrediction($logits_{base}$)
12:        **if** $pred_{base} == sample.label$ **then**
13:           $c_{base} \leftarrow c_{base} + 1$
14:        **end if**
15:        **// Step 2: Identification**
16:        $sk\_indices \leftarrow$ IdentifySK($M, inputs, layer$)
17:        $k \leftarrow |sk\_indices|$
18:        **if** $k = 0$ **then**
19:           **continue**
20:        **end if**
21:        $ctrl\_indices \leftarrow$ SelectRandomTokens($inputs, k,$ exclude $= sk\_indices$)
22:        **// Step 3: Intervention and Evaluation**
23:        $logits_{sk} \leftarrow$ Intervene($M, inputs, sk\_indices$)
24:        $pred_{sk} \leftarrow$ GetPrediction($logits_{sk}$)
25:        **if** $pred_{sk} == sample.label$ **then**
26:           $c_{sk} \leftarrow c_{sk} + 1$
27:        **end if**
28:        $logits_{ctrl} \leftarrow$ Intervene($M, inputs, ctrl\_indices$)
29:        $pred_{ctrl} \leftarrow$ GetPrediction($logits_{ctrl}$)
30:        **if** $pred_{ctrl} == sample.label$ **then**
31:           $c_{ctrl} \leftarrow c_{ctrl} + 1$
32:        **end if**
33:     **end for**
34:     $acc_{base} \leftarrow c_{base}/|D_{sub}|$
35:     $acc_{sk} \leftarrow c_{sk}/|D_{sub}|$
36:     $acc_{ctrl} \leftarrow c_{ctrl}/|D_{sub}|$
37:     Add $\{layer, acc_{base}, acc_{sk}, acc_{ctrl}\}$ to $R$
38: **end for**
39: **return** $R$

---

## C   DATASET AND PROMPT TEMPLATE

### C.1   SYSTEM PROMPT FOR BOOLQ

All experiments on the BoolQ dataset used the following System Prompt to instruct the LLMs. This prompt was designed to elicit a concise, binary response, aligning with the task's format.

```
You are a helpful assistant. Your task is to answer the following
   question with a single word: 'true' or 'false' based ONLY on
   the provided passage.
```

### C.2   DATA SAMPLES FROM BOOLQ

Below are three representative samples from the BoolQ dataset, formatted to illustrate the input structure provided to the models. The labels 'True' and 'False' correspond to the numerical values of 1 and -1 used during evaluation.

---

**Algorithm 2** Support Knowledge (SK) Identification

---

1: **Input:** Model $M$, Tokenizer $T$, Inputs $I$, Layer $l$
2: **Input:** Weights $\alpha, \beta$
3: **Output:** SK indices $S_{sk}$
4: **// Step 1: Get Model Activations**
5: $H, A \leftarrow M(I)$ (Get hidden states and attentions)
6: $h_l, a_l \leftarrow H_l, A_l$ (Select activations for layer $l$)
7: **// Step 2: Compute Component Scores**
8: $\Gamma_{grad} \leftarrow$ ComputeGradientNorms($M, I$)
9: $\Gamma_{attn} \leftarrow$ ComputeMaxMarginScores($h_l, a_l$)
10: **// Step 3: Combine and Select**
11: $\Gamma_{norm\_grad} \leftarrow$ Normalize($\Gamma_{grad}$)
12: $\Gamma_{norm\_attn} \leftarrow$ Normalize($\Gamma_{attn}$)
13: $\Gamma_{comb} \leftarrow \alpha \cdot \Gamma_{norm\_grad} + \beta \cdot \Gamma_{norm\_attn}$
14: $k^* \leftarrow$ ElbowMethod($\Gamma_{comb}$)
15: $S_{sk} \leftarrow$ TopKIndices($\Gamma_{comb}, k^*$)
16: **return** $S_{sk}$

---

**Algorithm 3** Mean-Replacement Intervention

---

1: **Input:** Model $M$, Inputs $I$, Indices to intervene $S_{intervene}$
2: **Input:** Intervention layer $l_{intervene}$
3: **Output:** Model logits after intervention $L_{intervened}$
4: $H \leftarrow M(I)$ (Get all hidden states)
5: $h_{target} \leftarrow H_{l_{intervene}}$ (Select hidden state at target layer)
6: **// Compute Mean Representation**
7: $\bar{h} \leftarrow \frac{1}{\text{seq\_len}} \sum_{t=1}^{\text{seq\_len}} h_{target}[t]$
8: **// Perform Replacement**
9: $h'_{target} \leftarrow h_{target}$
10: **for** each $idx \in S_{intervene}$ **do**
11:     $h'_{target}[idx] \leftarrow \bar{h}$
12: **end for**
13: **// Continue Forward Pass from Intervention Point**
14: $L_{intervened} \leftarrow$ ForwardPass($M, h'_{target}, \text{from\_layer} = l_{intervene}$)
15: **return** $L_{intervened}$

---

- **Sample 1:**
    - **Question:** can you marry a dead person in france
    - **Passage:** Posthumous marriage – Posthumous marriage (or necrogamy) is a marriage in which one of the participating members is deceased. It is legal in France and similar forms are practiced in Sudan and China. Since World War I, France has had hundreds of requests each year, of which many have been accepted.
    - **Label:** True

- **Sample 2:**
    - **Question:** is confectionary sugar the same as powdered sugar
    - **Passage:** Powdered sugar – Powdered sugar, also called confectioners' sugar, icing sugar, and icing cake, is a finely ground sugar produced by milling granulated sugar into a powdered state. It usually contains a small amount of anti-caking agent to prevent clumping and improve flow. Although most often produced in a factory, powdered sugar can also be made by processing ordinary granulated sugar in a coffee grinder, or by crushing it by hand in a mortar and pestle.
    - **Label:** True

- **Sample 3:**

- **Question:** is windows movie maker part of windows essentials
- **Passage:** Windows Movie Maker (formerly known as Windows Live Movie Maker in Windows 7) is a discontinued video editing software by Microsoft. It is a part of Windows Essentials software suite and offers the ability to create and edit videos as well as to publish them on OneDrive, Facebook, Vimeo, YouTube, and Flickr.
- **Label:** True

## C.3 SYSTEM PROMPT FOR RTE

All experiments on the RTE dataset used the following system prompt to instruct the LLMs. The prompt casts textual entailment as a binary decision over the labels `true` and `false`.

```
You are a natural language inference assistant. Given a premise
    and a
hypothesis, answer with a single token 'true' or 'false'. 'true'
    means
the premise entails the hypothesis; 'false' means the hypothesis
    is not
entailed or contradicts the premise. Output only 'true' or 'false
    '.
```

## C.4 DATA SAMPLES FROM RTE

Below are three representative samples from the RTE dataset, formatted as premise–hypothesis pairs. The label 1 denotes entailment (mapped to the token `true`), while the label 0 denotes non-entailment or contradiction (mapped to `false`).

- **Sample 1:**
    - **Premise:** Amazon shares fell nearly 4 percent following the results as the company said operating income would drop as much as 42 percent in the second quarter.
    - **Hypothesis:** Shares of Amazon fell 4 percent.
    - **Label:** 1 (entails)

- **Sample 2:**
    - **Premise:** Traditionally, the Brahui of the Raisani tribe are in charge of the law and order situation through the Pass area. This tribe is still living in present day Balochistan in Pakistan.
    - **Hypothesis:** The Raisani tribe resides in Pakistan.
    - **Label:** 0 (not entailed / contradicts)

- **Sample 3:**
    - **Premise:** In Nigeria, by far the most populous country in sub-Saharan Africa, over 2.7 million people are infected with HIV.
    - **Hypothesis:** 2.7 percent of the people infected with HIV live in Africa.
    - **Label:** 1 (entails)

## C.5 SYSTEM PROMPT FOR CNN/DAILYMAIL

For CNN/DailyMail, we treat summarization as a single-paragraph generation task. The following system prompt instructs the model to produce concise, factual news summaries.

```
You are a news summarization assistant. Given a news article,
    write one
concise, factual paragraph that captures only the key information.
Do not provide step-by-step reasoning or internal thoughts.
Output only the final summary, without any
```

```
<div class="think">...</div>

 content.
Write in clear journalistic style, using 1-3 sentences and at most
    about
60 English words, avoiding repetition, dialogue and rhetorical
    questions.
```

### C.6 DATA SAMPLES FROM CNN/DAILYMAIL

Below are three representative samples from the CNN/DailyMail dataset, each consisting of a news article and its reference highlights.

- **Sample 1:**
  - **Article:** (CNN) A shooting at a bar popular with expatriates in Mali on Saturday killed five people, including French and Belgian citizens, authorities said. One French citizen, one Belgian and three Malians were killed in the attack in the capital of Bamako, said Gabriel Toure, ...
  - **Highlights:** A jihadist group claims responsibility in an audio recording, news agency reports. The Malian government calls the shooting a "terrorist act". One French citizen, one Belgian and three Malians are killed.

- **Sample 2:**
  - **Article:** (CNN) On the 6th of April 1996, San Jose Clash and DC United strode out in front of 31,683 expectant fans at the Spartan Stadium in San Jose, California. The historic occasion was the first ever Major League Soccer match – a brave new dawn for the world's favorite sport in a land its charms had yet to conquer. Summarizing the action for ESPN, ...
  - **Highlights:** The 20th MLS season begins this weekend. League has changed dramatically since its inception in 1996. Some question whether rules regarding salary caps and transfers need to change.

- **Sample 3:**
  - **Article:** (CNN) A Pennsylvania community is pulling together to search for an eighth-grade student who has been missing since Wednesday. The search has drawn hundreds of volunteers on foot and online. The parents of Cayman Naib, 13, ...
  - **Highlights:** Cayman Naib, 13, hasn't been heard from since Wednesday. Police, family, volunteers search for the eighth-grader.

## D DETAILED ANALYSIS OF INTERVENTION EXPERIMENTS

Figure 5 presents the intervention results across a diverse set of LLMs. While all models confirm the central finding that intervening on the neural units associated with SK tokens induces a far greater performance loss than random interventions, the cross-model comparison allows for a deeper analysis of architectural and scaling effects.

We observe distinct patterns in the stability and distribution of layer-wise sensitivity. For instance, the accuracy curve for Llama3.1-8B under intervention on SK-related neural units displays significant volatility, with sharp drops in middle layers followed by recovery. In contrast, the Qwen3 family models exhibit a more stable plateau of low accuracy across their mid-to-late layers. This suggests that while all models rely on the information encoded in these units, their dependency might be concentrated in sharp, critical junctures in some architectures, while being more sustained and distributed across a broader decision-making region in others.

Furthermore, examining models of varying sizes within the Qwen3 family reveals a scaling trend. For smaller LLMs (8B, 14B), this optimal intervention layer appears in the early stages, whereas

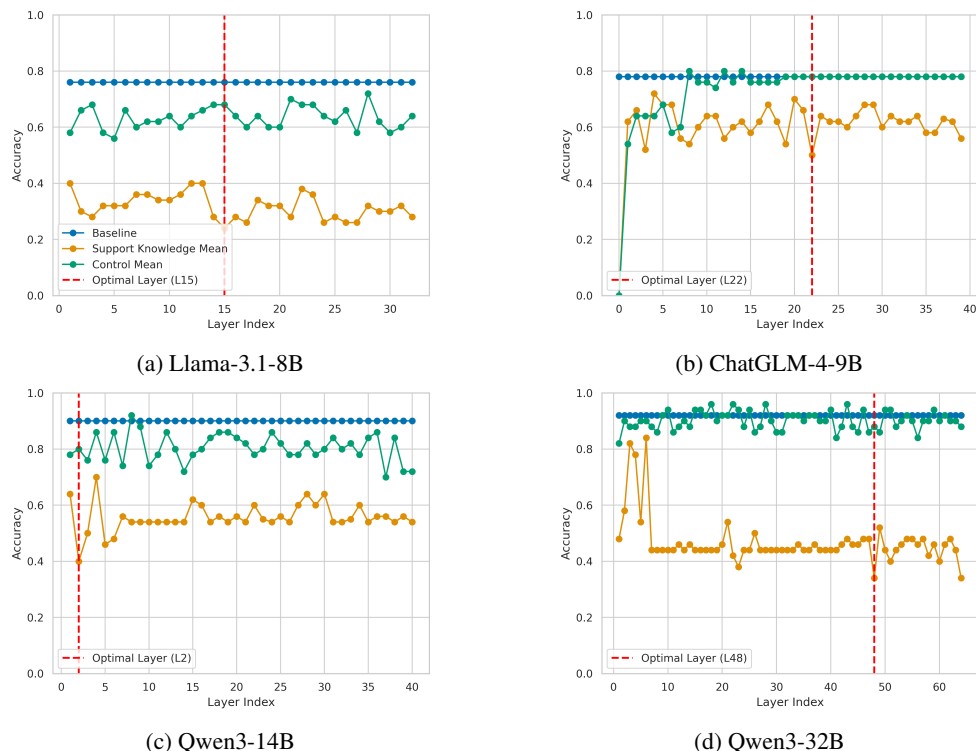

Figure 5: Detailed layer-wise impact of causal interventions across models. This figure provides the full layer-wise breakdown for the intervention experiments summarized in Figure 4 of the main text. For each model, Support Knowledge tokens were identified at each layer, followed by a mean-replacement intervention on the corresponding tokens at the embedding layer. The consistent accuracy drop after intervening on Support Knowledge tokens (orange line), compared to the baseline (blue line) and random control (green line).

for the 32B model, it shifts to the mid-to-late layers. This posterior shift, combined with the greater stability of the accuracy drop in larger models, suggests that as model capacity increases, the critical reasoning steps become more consolidated and are deferred to more abstract, later stages of processing.

# E  DETAILED ANALYSIS OF SUPPORT KNOWLEDGE EVOLUTION IN LLAMA3.1-8B

This section provides a detailed layer-by-layer analysis of the Support Knowledge evolution for the representative sample processed by Llama3.1-8B. This analysis offers a granular view that complements the aggregate patterns discussed in the main paper, providing a micro-level illustration of the model's two-phase reasoning process.

In the initial layers (1-7), the model engages in a broad exploratory phase. The SK set is large and diverse, reflecting a comprehensive scan of the entire input. For instance, at Layer 1, the SK set contains 70 tokens, including a wide array of structural markers such as `<|begin_of_text|>`, `<|end_header_id|>`, and task-framing words like "assistant" and "user". Crucially, it also includes content-specific words from the passage, such as "interstate" and "coast", alongside potential answer tokens "true" and "false". This initial breadth suggests the model has not yet prioritized information, instead treating all parts of the prompt as potentially relevant.

A significant shift occurs in the middle layers, a behavior consistent with the aggregate analysis in the main paper where Llama3.1-8B was shown to concentrate its evidence evaluation in this specific region. The model begins to consolidate its understanding and focus on core evidence. While the

number of SKs fluctuates, the composition of the set becomes more refined. The structural markers remain, but their dominance wanes as the model increasingly hones in on task-critical information. For example, at Layer 14, while the total number of SKs is 63, the set now strongly features key evidence tokens like "longest", "coast-to-coast", and "Interstate 90", alongside the definitive answer pair "true"/"false". This period marks the active evidence evaluation phase, where the model weighs the information extracted from the passage against the task's binary choice requirement.

In the final layers (approximately 25-32), the SK distribution enters a stable phase, locking onto a minimal, highly influential set of tokens. The size of the SK set remains consistently around 55 tokens, but its composition is now highly focused. The persistent SKs in these deep layers almost exclusively consist of the core task-defining elements (e.g., "single word", "true", "false") and the most critical pieces of evidence (e.g., "interstate", "coast"). The initial exploratory tokens from the early layers have been almost entirely pruned. This final, stable set represents the model's converged internal representation—a sparse, causally critical foundation upon which its final decision is made. This detailed evolution from a wide, exploratory set to a narrow, decisive one provides a clear micro-level illustration of the two-stage computational strategy.

## F    DETAILED ANALYSIS OF SUPPORT KNOWLEDGE EVOLUTION IN CHATGLM4-9B

This section provides a detailed layer-by-layer analysis of the Support Knowledge evolution for the representative sample processed by the ChatGLM4-9B model. This trace complements the aggregate analysis in the main paper, which identified ChatGLM4-9B's strategy of postponing its primary evidence evaluation to the final layers.

For the majority of the model's depth (layers 1-31), ChatGLM4-9B operates in a prolonged task construction phase. The SK set converges to a minimal group of tokens and remains consistent. From Layer 4 onwards, the number of SKs is approximately 20. An examination of these SKs reveals that they are almost exclusively structural and task-defining. For instance, at Layer 10, the set is composed of prompt elements such as `<|assistant|>`, `<|user|>`, `<|system|>`, and template words like "question", "passage", "true", and "false". Content-specific words from the passage, such as "interstate" or "coast", are absent during this phase. This indicates the model spends most of its computational effort processing the task's structure and requirements, while deferring the integration of specific evidence.

The second phase, evidence evaluation, occurs in the final layers (32-40). The shift is marked by a clear change in SK composition. At Layer 32, the number of SKs increases from 18 to 41. This expanded set now includes content-specific knowledge from the passage for the first time, such as "parallels", "Northwest", "U.S.", and "designation". This pattern persists through the final layers, where the model weighs this newly integrated evidence against the established task frame to arrive at its decision. For example, at the final layer (Layer 40), the SK set still contains these evidence tokens alongside the core task-framing elements.

This detailed trace for ChatGLM4-9B illustrates its computational timeline. The model's strategy—an extended task construction phase followed by a compressed, late-stage evidence evaluation—differs from other architectures and reinforces the idea that while the two-phase process is a general principle, its implementation can be model-specific.

## G    DETAILED ANALYSIS OF SUPPORT KNOWLEDGE EVOLUTION IN QWEN3-14B

This section provides a detailed layer-by-layer analysis of the Support Knowledge evolution for the representative sample processed by the Qwen3-14B model. The trace complements the aggregate analysis in the main paper, which characterized the Qwen family's dependency on SK as a sustained and distributed process.

The evolution in Qwen3-14B presents a distinct variant of the two-phase model. In the initial layers (1-7), the model performs its task construction. The SK set is at its most diverse during this phase, containing a mix of structural markers (`<|im_start|>`, `</think>`), task-defining words

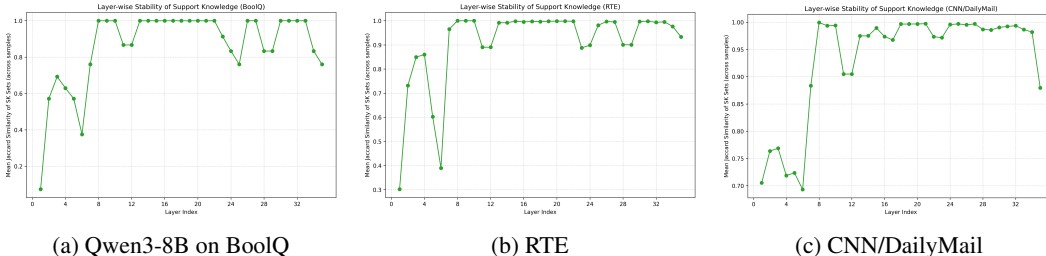

(a) Qwen3-8B on BoolQ      (b) RTE      (c) CNN/DailyMail

Figure 6: Adjacent-layer Jaccard overlap between SK sets across models and tasks. Each curve shows, for a given model and dataset, the Jaccard similarity between SK sets in consecutive layers. All three tasks display a two-segment pattern: low and unstable overlap in shallow layers, followed by a high-overlap plateau in deeper layers, indicating that SK is repeatedly reconfigured in the first segment and then locked into a stable configuration in the second segment.

("Question", "true", "false"), and content-specific evidence from the passage ("interstate", "coast", "Northwest"). For example, at Layer 2, the 45 identified SKs are spread across all sections of the prompt, indicating a broad initial information scan.

Qwen3-14B transitions into a prolonged phase of sustained refinement from approximately Layer 8 onwards. The number of SKs remains highly consistent, averaging around 47 for the remainder of the model's depth. More importantly, the composition of the SK set also stabilizes early. Core evidence tokens such as "interstate", "coast-to-coast", and "stretches", first identified in the initial layers, are retained as persistent SKs through to the final layer. Similarly, key task-framing tokens like "true", "false", and "ONLY" are locked in for the duration. For instance, the SK set at Layer 10 is nearly identical to the set at Layer 38.

This trace reveals the model's specific computational strategy. Instead of deferring evidence integration, Qwen3-14B appears to identify a comprehensive set of causally critical tokens early in its processing. The subsequent layers then operate continuously on this stabilized set, refining the relationships between the established task frame and the core evidence. This "early identification, sustained refinement" approach aligns with the observation from the main paper's intervention experiments, where the impact of SK neutralization manifests as a stable plateau across the model's mid-to-late layers, rather than a sharp peak. This confirms that while the two-phase principle is general, its implementation timeline is a key architectural differentiator.

## H DETAILED ANALYSIS OF SUPPORT KNOWLEDGE EVOLUTION IN QWEN3-32B

This section provides a layer-by-layer analysis of the Support Knowledge evolution for the representative sample processed by the Qwen3-32B model. This trace complements the aggregate analysis from the main paper, which identified a posterior shift in causal sensitivity and a more stable dependency pattern in larger models of the Qwen family.

Similar to the Qwen3-14B model, the evolution in Qwen3-32B is characterized by an "early identification, sustained refinement" strategy, though executed with even greater speed and stability. In the earliest layers (1-6), the model performs a swift pruning of its initial broad set of SKs. For instance, the SK set size decreases from 62 tokens at Layer 1 to 31 tokens by Layer 6. This initial set contains a wide range of tokens, but the model quickly filters out a significant portion.

From Layer 7 onwards, the model enters a remarkably stable and extended phase of sustained refinement that persists for nearly 60 layers. The number of SKs locks at approximately 38 for the remainder of the model's depth. The composition of this set is also highly consistent. Key task-defining tokens such as "assistant", "Question", "true", "false", and "ONLY", are identified early and remain in the SK set throughout. Similarly, critical evidence from the passage, including "interstate", "coast", "Idaho", "Yellowstone", and "Highway", are also part of this persistent set. For example, the 38 tokens identified as SKs at Layer 10 are almost identical to those at Layer 60.

This trace illustrates the specific computational strategy of the Qwen3-32B model. It quickly converges on a minimal, causally critical set of information and then dedicates the vast majority of its layers to processing this fixed set. This explains the stable plateau of intervention impact observed in the main paper. Furthermore, by locking onto the core information early, the model can engage in more extended, abstract processing in its deeper layers. This aligns with the finding that the most causally sensitive layers in larger models tend to shift towards the posterior, as the critical reasoning steps are deferred to these later stages of refinement.

## I    QUANTITATIVE ANALYSIS OF THE TWO-SEGMENT MECHANISM

In this section we provide a quantitative view of the two-segment inference mechanism using layer-wise similarity metrics. Figure 6a reports the adjacent-layer Jaccard overlap between SK sets for Qwen3-8B on BoolQ, while Figures 6b and 6c show the same statistic for RTE and CNN/DailyMail, respectively. At each layer, SK tokens are first identified using the elbow method described in Equation 8, and we then compute the Jaccard similarity between the SK sets of consecutive layers.

Across all three tasks, the curves exhibit a clear two-segment pattern. In the shallow layers (approximately 1–7), the adjacent-layer Jaccard overlap is relatively low and volatile, indicating that the model is actively reshuffling which tokens are treated as SK. In contrast, from the middle layers onward the overlap rapidly increases and stays close to 1.0, with many layers achieving an exact match between successive SK sets. This high-overlap plateau shows that once the model enters the second segment, it effectively locks onto a stable configuration of SK tokens and reuses this compact set of representations throughout the remaining depth.

## J    FUTURE DIRECTION: A FRAMEWORK FOR SALIENCY-ALIGNED DISTILLATION

As briefly mentioned in the main paper's Discussion, our Support Knowledge (SK) framework offers a potential pathway to refine existing knowledge distillation techniques. Traditional distillation methods typically treat the teacher model as a black box, compelling the student to mimic output distributions (e.g., logits) without understanding which specific input elements were critical to the teacher's conclusion. This section elaborates on a conceptual framework for a more transparent and targeted form of distillation, which we term saliency-aligned distillation.

Instead of merely matching outputs, the core idea is to guide the student model to replicate the teacher's internal attentional focus on the same causally important tokens. This can be implemented through an auxiliary loss term that encourages alignment between the teacher's and student's Support Knowledge distributions:

$$\mathcal{L}_{\text{SK}} = D_{\text{KL}}(P_T \parallel P_S) = \sum_{t=1}^{n} P_{T,t} \cdot \log\left(\frac{P_{T,t}}{P_{S,t}}\right) \tag{9}$$

where $P_T = \text{softmax}(\Gamma_T/\tau)$ and $P_S = \text{softmax}(\Gamma_S/\tau)$ represent the normalized SK scores for the teacher and student models, respectively, and $\tau$ is a temperature parameter. This loss term effectively penalizes the student for focusing on information the teacher deems irrelevant, while rewarding it for attending to the same critical evidence.

The final distillation objective would then combine this saliency-alignment loss with the traditional task-specific loss:

$$\mathcal{L}_{\text{total}} = (1 - \lambda) \cdot \mathcal{L}_{\text{distill}} + \lambda \cdot \mathcal{L}_{\text{SK}} \tag{10}$$

Here, $\mathcal{L}_{\text{distill}}$ can be the standard Kullback-Leibler divergence loss between the teacher's and student's output logits. We hypothesize that such an approach could yield more sample-efficient and generalizable student models that are also inherently more interpretable, due to their aligned internal salience patterns.

While a full implementation and empirical validation are beyond the scope of this paper, saliency-aligned distillation represents a promising avenue for future research, extending our Support Knowledge framework toward practical applications.

## K    COMPUTATIONAL INFRASTRUCTURE

All experiments were conducted on a single server equipped with the hardware and software detailed below.

### K.1    HARDWARE SPECIFICATIONS

The computational experiments were run on a machine with the following hardware configuration:

- **GPU:** 1x NVIDIA RTX A6000 (48 GB VRAM)
- **CPU:** Intel(R) Xeon(R) Gold 6133 CPU @ 2.50GHz
- **System RAM:** 64 GB

### K.2    SOFTWARE ENVIRONMENT

The software environment was managed using Python. The core dependencies required to run our analysis scripts, particularly for Support Knowledge identification, are listed below. We also provide the specific versions used in our experiments for exact reproducibility.

- **Python Version:** 3.10.12
- **Core Dependencies (Minimum Versions):**
    - `torch >= 2.0.0`
    - `transformers >= 4.30.0`
    - `numpy >= 1.21.0`
- **Specific Versions Used for Experiments:**
    - `torch == 2.2.2`
    - `transformers == 4.44.0`
    - `numpy == 1.24.3`
    - `CUDA Version:` 12.1

