# OpenReview forum: "Identifying Support Knowledge Representation for Large Language Models"
_ICLR.cc/2026/Conference — Submitted to ICLR 2026_

### Official Review · Reviewer_NE1S · 2025-10-26

**Soundness:** 2
**Presentation:** 3
**Contribution:** 2
**Rating:** 2
**Confidence:** 3

**Summary:**

This paper introduces and systematically investigates the concept of Support Knowledge (SK), proposing that LLMs rely on a small, causally critical set of atomic tokens during reasoning analogous to support vectors in SVMs. To identify these tokens, the authors present a Gradient-Attention framework that combines two complementary  gradient  and attention to dynamically locate the input tokens the model truly depends on.

The findings are intuitive and supported by thorough experiments. However, the work stops short of demonstrating practical gains from SK. Future efforts that integrate SK into model training or optimization could significantly enhance its practical relevance and impact.

**Strengths:**

1. Comprehensive analysis: The paper conducts extensive evaluations across multiple models (Qwen3, Llama3.1, ChatGLM4) and scales (8B–32B), showing that the proposed Support Knowledge (SK) phenomenon consistently exists across architectures.

    It further includes layer-wise analysis, cross-model comparisons, ablation studies, and baseline evaluations, which together make the conclusions convincing and well-grounded.

2. interpretability: The identified SK tokens often carry clear semantic meaning (e.g., “true”, “interstate”), intuitively reflecting the model’s reasoning logic. Moreover, the visualization of SK evolution across layers provides a clear and interpretable view of how reasoning dynamics emerge within the model.

**Weaknesses:**

1. Limited task coverage

All experiments are conducted on BoolQ, a binary QA task. It remains unclear whether the proposed findings generalize to other task types such as reasoning, summarization, or multi-turn dialogue.
The sparsity assumption of SK also remains untested in open-ended generation tasks (e.g., text or code generation), where token dependencies may differ substantially.

2. Lack of downstream validation (my core concern)

- Although the paper effectively identifies SK tokens, it does not apply them in fine-tuning, preference optimization, or reinforcement learning to assess their practical utility.
- As a result, SK currently functions as an interpretability tool rather than an optimization signal. Demonstrating its value in improving model performance or training efficiency would make the work more impactful.

**Questions:**

Did the authors use SK tokens for fine-tuning or reinforcement learning to verify their practical utility? The paper primarily focuses on interpretability and intervention experiments, rather than applying SK as a training signal. Although the idea of SK-aligned distillation is introduced, it has not been implemented or empirically validated. As it stands, SK remains an analytical concept rather than a demonstrated optimization method.

Related studies have explored similar directions, for instance, phi-4[1] identifies key tokens through multiple sampling for DPO, while [2] uses high-entropy tokens to stabilize RL training. It would be interesting to see whether the authors could leverage SK-identified tokens for comparable downstream applications, such as fine-tuning, preference optimization, or reinforcement learning.

[1] Abdin M, Aneja J, Behl H, et al. Phi-4 technical report[J]. arXiv preprint arXiv:2412.08905, 2024.

[2] Wang S, Yu L, Gao C, et al. Beyond the 80/20 rule: High-entropy minority tokens drive effective reinforcement learning for llm reasoning[J]. arXiv preprint arXiv:2506.01939, 2025.

---

> ### Author Response · Authors · 2025-11-24
>
> We thank you for the thoughtful feedback and for appreciating our comprehensive analysis. However, we believe that a rating of "2: reject, not good enough" may understate the strength and significance of our foundational validation experiments. We clarify as follows:
>
> **1. On Causal Intervention as Foundational Evidence**
>
> We agree that downstream validation is essential for demonstrating the practical impact of any explanation method. In our work, the intervention experiments are designed to provide the most direct form of evidence available in our setting: they test whether the tokens selected by our method are causally necessary for the model’s predictions.
>
> As shown in Table 1 of the revised manuscript (Sec. 4.3), ablating the identified Support Knowledge (SK) tokens leads to a dramatic drop in model accuracy (from 82.34% to 27.78%), whereas ablating tokens selected by other heuristics yields substantially smaller drops. This pattern indicates that SK captures a compact subset of tokens to which the model’s decisions are highly sensitive. In this sense, our analysis goes beyond purely observational correlations and provides strong evidence that SK tokens exert a causal influence on BoolQ performance. In addition, Sec. 4.2.2 and Appendix Sec. I extend these intervention-based comparisons, showing that the same qualitative pattern of SK-targeted interventions outperforming baselines also holds on RTE and CNN/DailyMail. We fully agree that an important direction for future work is to turn this insight into explicit training objectives, such as auxiliary SK‑based distillation losses that encourage a student model to match the teacher’s focus on SK tokens (see Appendix Sec. J).
>
> **2. On the Potential of SK as an Optimization Signal**
>
> While we did not implement a full fine-tuning or RL experiment in this submission, we agree that turning SK into a training signal is a natural next step. In the revised manuscript, the Conclusion sections (Sec.5) explicitly frame SK as a diagnostic tool for understanding model behavior, and Appendix Sec. J ("Future Direction: A Framework for Saliency-Aligned Distillation") sketches a possible SK-aligned distillation objective. There, we explicitly present this objective as a conceptual blueprint rather than an experimentally validated method.
>
> We also thank you for pointing to the excellent work on Phi-4 (Abdin et al., 2024) and on high-entropy minority tokens for LLM reasoning (Wang et al., 2025). These works convincingly demonstrate that identifying critical tokens can benefit optimization. Our contribution is complementary: our Gradient–Attention framework provides a mechanistic procedure for locating a sparse set of support tokens, grounded in the model’s internal computation (gradients and information flow via attention), rather than in sampling or entropy heuristics.
>
> In principle, such SK tokens could be plugged into existing optimization frameworks. For example, in Direct Preference Optimization (DPO), one might restrict rewards to SK tokens so that learning focuses on the parts of the response that carry the core reasoning signal. Exploring whether SK-based signals actually improve sample efficiency or generalization, however, requires substantial additional experimentation and is beyond the scope of this paper. We will explicitly position these ideas as promising directions for future work, not as claims established by our current experiments.
>
> **3. On the Scope of Task Coverage**
>
> We acknowledge that our primary experiments are conducted on the BoolQ dataset. This choice was deliberate: its binary QA format allows for clean and unambiguous causal interventions and evaluation via accuracy. In the revised manuscript, we further add supplementary experiments on RTE (textual entailment) and CNN/DailyMail (long-document summarization) in Sec. 4.1, Sec. 4.2.2, Table 1, and Appendix Sec. I. These additional results show that (i) SK remains sparse, (ii) the two-segment inference mechanism also appears beyond BoolQ, and (iii) SK-targeted interventions continue to produce the largest performance degradations relative to random, positional, and single-signal baselines across tasks.

---

> > ### Comment · Reviewer_NE1S · 2025-11-25
> >
> > Thank you for the additional experiments. However, my core concern remains unresolved, and I maintain my score. The authors acknowledge: "While we did not implement a full fine-tuning or RL experiment..." Intervention experiments show SK tokens are important for predictions, but it remains unclear whether they can improve training—these are different questions. Related works (Phi-4, high-entropy tokens) identify important tokens and use them to improve models via DPO and RL. Positioning SK only as a "diagnostic tool" with optimization deferred to future work is insufficient.
> >
> > In my opinion, identifying special tokens must demonstrate downstream utility, purely diagnostic tools cannot establish practical value. The authors have the framework; adding one simple experiment (SK-weighted loss, selective fine-tuning, or data filtering) would validate whether SK tokens can guide optimization. Without this, SK remains theoretical without demonstrated impact.

---

> > > ### Author Response · Authors · 2025-12-04
> > > **Official Comment by Authors**
> > >
> > > To directly address your request for a training-time use of SK tokens, we additionally ran SK‑weighted fine‑tuning experiments on the BoolQ dataset (9,427 train / 3,270 dev / 3,245 test examples) using Qwen3‑8B, with baseline and SK‑weighted runs sharing same hyperparameters. In our experiments, we set the loss weight of SK tokens to 4. The results show that the  dev accuracy improves from 84.6% (baseline) to 85.7% (SK‑weighted), corresponding to an absolute gain of about 1.1%. This provides evidence that SK tokens can supply useful gradient signal for optimization rather than being purely diagnostic.
> > >
> > > For completeness, we also applied the same procedure to the smaller RTE dataset (2,500 train / 278 dev / 300 test examples). There, the SK‑weighted models improved accuracies slighty (89.53% for SK-weighted and 89.17% for baseline).

---

### Official Review · Reviewer_Wss1 · 2025-10-29

**Soundness:** 2
**Presentation:** 4
**Contribution:** 2
**Rating:** 4
**Confidence:** 3

**Summary:**

The paper introduces Support Knowledge (SK), a sparse set of “causally critical” tokens identified by a combined gradient–attention score. The paper argues that Transformer inference unfolds in a two-segment pattern: early evidence gathering and later task framing. Concretely, the method computes per-token gradient norms and an attention-based max-margin term, normalizes them, mixes them with model-specific weights $\alpha$, $\beta$($\alpha+\beta=1$) , then selects a sparse set via an elbow rule; causal importance is probed by mean-replacement ablations of hidden states at a chosen layer. On BoolQ, targeting SK tokens yields substantially larger accuracy drops than random or simple baselines; similar qualitative patterns are reported for Llama3.1, ChatGLM4, and Qwen3.

**Strengths:**

- **Clear writing**: The writing is well-organized, moving from theoretical motivation to method and experiments with helpful figures and algorithms.

- **Simple, testable methodology**: Concrete SK definition, elbow-based selection, and mean-replacement interventions are spelled out with algorithms, aiding reproducibility.

- **Strong experiment result**: Reported drops are large under the chosen intervention (e.g., substantial degradation when ablating only a 17% SK subset), suggesting the score captures influential tokens.

- **Empirical patterning across models**: Reports of the two-segment SK distribution and layer-wise ablation sensitivity across multiple LLM families.

**Weaknesses:**

- Theory seems to be ad-hoc, The connection between LLM and SVM has been established before. Most of the theory part in the paper appears to be an ad-hoc. And the connection between attention gradient and SVM is too far-fetched.
- A lot of methods identifying important tokens exist[1][2][3]. To my knowledge, the key contribution of the paper seems to be identifying important tokens easily. However, the SK methods is not straightforward, requiring tuning $\alpha$ and $\beta$. ($\alpha+\beta=1.1$ for LlaMA in Table 2, contradiction of paper's setting).

[1] Chefer, Hila, Shir Gur, and Lior Wolf. "Transformer interpretability beyond attention visualization." In Proceedings of the IEEE/CVF conference on computer vision and pattern recognition, pp. 782-791. 2021.
[2] Bai, Yu, Heyan Huang, Cesare Spinoso-Di Piano, Marc-Antoine Rondeau, Sanxing Chen, Yang Gao, and Jackie Chi Kit Cheung. "Identifying and Analyzing Performance-Critical Tokens in Large Language Models." arXiv e-prints (2024): arXiv-2401.
[3] Meng, Kevin, David Bau, Alex Andonian, and Yonatan Belinkov. "Locating and editing factual associations in gpt." Advances in neural information processing systems 35 (2022): 17359-17372.

**Questions:**

- Theory
  - The subscript $x_t \in T$ in equation (4) should be $t \in [1,T] \cap Z$. What is the relationship between equation (4) and the following content?
  - They state $||\nabla_{x_t} f(X)|| \propto \partial L / \partial d_t$, $d_t$ ​is the “distance to the decision boundary manifold,” and conclude tokens with large gradients are “like support vectors.” But the paper doesn’t define $d_t$ operationally for LLM. The boundary/manifold framing is imported from SVMs without a workable definition here.
  - Theorem 1 is very elegent, but what are definitions of $\alpha$, $b$ here? Where is the proof? How to map Transformer objectives/constraints to an SVM Lagrangian? What is the exact formulation of your primal and dual problem?
- Experiment and Method
  - In Definition 1, '$a_t$ is the attention weight on token $t$'. However, attention is a matrix. Is this attention weight between predicted token and token $t$.
  - Most of your competitors in Table 1 are generally weak. Have you compare with stronger competitors? And your setting, mean-replacment invention seems to be unconventional. It appears that you create a personal benchmark and compare with weak baselines.

In general, the paper presents intriguing empirical patterns and a practical scoring rule. However, I personally does not get the meaning of the method, as a lot of methods identifying important tokens already exist. And the theory seems to be rather ad-hoc. I don't fully grasp it and welcome other reviewers' insights.

---

> ### Author Response · Authors · 2025-11-24
> **Official Comment By Author （Part1）**
>
> We sincerely thank you for the thorough review and insightful feedback. We appreciate the positive comments on our paper's presentation and the strength of our empirical results.
>
> **1. On the Theoretical Foundation**
>
> - In revision, Sec. 3.2 now presents the SVM connection purely as an analogy at the level of linear scoring and primal–dual structure. We emphasize that attention scores and gradients are interpreted as **dual-like** coefficients for identifying influential tokens, without claiming that backpropagation literally solves an SVM dual problem.
> - The gradient–distance relation is now explicitly framed as a first-order Taylor approximation (Eq. (7)). We explain that $d_t$ is a scalar coordinate parameterizing distance to a local decision boundary manifold, used only to motivate a local sensitivity relation $\|\nabla_{x_t} f(X)\| \propto \partial \ell / \partial d_t$.
> - We revise Definition 1 to state that $a_t$ denotes the attention weight on token $t$ in the layer where the SK score is computed. We also simplified Eq. (5) by removing redundant terms to improve clarity, and corrected minor typos and indexing issues reported in the review.
>
> Our main contribution is empirical: the discovery and causal validation of SK and its two-segment inference mechanism, rather than a new closed-form SVM theory.
>
> **2. On Contribution and Novelty**
> Our work differs from prior methods in goal, analytical perspective, and primary findings.
>
> * **Chefer et al. (2021):** Chefer et al. aim to attribute the final prediction to all input tokens via Layer-wise Relevance Propagation (LRP), producing a dense relevance map. Our goal is different: we study a sparse subset of tokens—Support Knowledge (SK)—whose associated representations are empirically shown to be causally important for the model’s predictions. While their method is a post-hoc relevance assignment, our framework is conceptually motivated by margin-based views (such as support vectors in SVMs) and focuses on analyzing how SK behaves across layers. Accordingly, their primary contribution is an attribution method, whereas we provide an empirical characterization of SK and its two-segment inference pattern.
>
> * **Bai et al. (2024):** Bai et al. investigate In-Context Learning (ICL) via macro, input-level ablations of pre-defined token categories (e.g., templates vs. content) and show that template tokens are important for ICL task formats. Our work, in contrast, focuses on zero-shot reasoning and performs a micro, layer-wise internal analysis. We propose a dynamic SK scoring mechanism and track how SK tokens change across layers. Our experiments suggest that, in zero-shot reasoning, SK tokens follow a two-segment pattern, transitioning from an evidence-gathering phase to a task-framing phase.
>
> * **Meng et al. (2022):** These works address different questions. Meng et al. (ROME) ask “Where is factual knowledge stored, and how can it be edited?” and localize factual associations within mid-layer MLPs, together with an editing mechanism. Our work instead asks “How does a model reason using its internal knowledge?” Our contribution is not a new editing technique, but an interpretability framework that analyzes a sparse SK structure that appears to underpin part of the model’s reasoning process.

---

> ### Author Response · Authors · 2025-11-24
> **Official Comment By Author (Part 2)**
>
> **3. On Experimental Design**
>
> - **Baselines and mean-replacement (Table 1).** In order to clarify your concern that our baselines and the mean-replacement intervention in Table 1 might be unconventional, we revise Sec. 4.3 and the description of Table 1 to more explicitly define each baseline and the intervention protocol. Table 1 is designed to test whether SK tokens are more causally important than tokens selected by simple baselines. We compare SK-based interventions to random, positional, pure-attention, and pure-gradient baselines across BoolQ, RTE, and CNN/DailyMail (Sec. 4.3, Table 1). Mean-replacement at the hidden state level follows standard causal intervention practice in knowledge-editing work (e.g., Meng et al., 2022): it removes token-specific information while preserving the overall distribution. Across all tasks, SK-targeted interventions consistently cause the largest performance degradation, indicating that SK tokens form the most causally critical subset.
> - **Model-specific weights and attention weights.** In order to clarify the reviewer's question about how the model-specific weights ($α, β$) and the attention weights ($a_t$) are defined in the SK score, we revise Appendix Sec. “Model-Specific Weighting for SK Score Calculation” and the text around Definition 1 to make these choices explicit. Appendix Sec. "Model-Specific Weighting for SK Score Calculation" reports the calibrated $(α, β)$ for each model (Table 2) and explains that we choose them to maximize intervention impact at the most sensitive layer, revealing model-dependent reliance on gradients vs. attention. We corrected the clerical error for Llama in the previous table (now $α=0.7, β=0.3$). In the revised text around Definition 1, we clarify that $a_t$ is the attention weight on token $t$ in the layer where the SK score is computed, so that the SK score is grounded in the actual attention pattern used by the model.

---

> ### Comment · Reviewer_Wss1 · 2025-11-26
>
> **Theory remains a conern** \
> As I've pointed out theory has a lot of typo, some are fixed, some persist
> - "$a_t$ denotes the attention weight on token $t$ in the layer where the SK score is computed." Again, attention is a matrix, It should be attention between two tokens!
> - $||\nabla_{x_t} f(X)|| \propto \partial L / \partial d_t$, $d_t$ ​is the “distance to the decision boundary manifold,” and conclude tokens with large gradients are “like support vectors.” Where is your definition of boundary manifold and support vector, where is your proof? **The most extradionary thing for this work is that It neither has clear definition nor proof, even after rebuttal.**
>
> **Lack of application value**
> > Our main contribution is empirical: the discovery and causal validation of SK and its two-segment inference mechanism, rather than a new closed-form SVM theory.
>
> You classified important token (Support Knowledge token). What can we get from important tokens? Do you get any novel explanation for model mechanism? Is there any application value demonstrated? I agree with Reviewer NE1S in this way.

---

### Official Review · Reviewer_KSdf · 2025-10-30

**Soundness:** 3
**Presentation:** 3
**Contribution:** 3
**Rating:** 6
**Confidence:** 2

**Summary:**

The paper proposes a novel method for identifying tokens in the attention mechanism which are important to the model's output in a classification task. They show that "masking" these tokens results in drastic drop in performance, compared to masking a tokens at random. The proposed method shows interesting results in terms of LLM interpretability on the BoolQ dataset and seems applicable to other tasks, which should be of interest to the community.

**Strengths:**

- Interesting findings.
- Useful seemingly novel method for LLM interpretability.

**Weaknesses:**

While the findings are interesting, the authors only explored a classification task, namely the BoolQ dataset. While it would be nice to have results on more tasks, I believe that the findings are already sufficient. My problem is with how certain statements in the discussion section are phrased. For instance, they write "Our findings indicate that LLMs do not learn in a uniformly distributed manner, but instead converge
on a sparse, structured computational principle we term Support Knowledge (SK)." This is only true for the BoolQ dataset. The authors do however state right after that "While this study focuses on the binary QA format to cleanly isolate these core mechanisms, future work should explore their generalizability to other tasks." So I do not believe that the authors are intentionally over-claiming, but the way that certain sentences are phrased gives that impression. This is a relatively minor issue, but the paper would be better in my opinion if such claims were better phrased/contextualized.

**Questions:**

- You use the term "machanism" throughout the paper. Is this a typo or is this actually a word?
- When referring to the Appendix, you do not refer to the section in the Appendix.
- Line 29-30, I am not sure what this sentence is conveying, are you sure that the syntax is correct?
- Line 35-36 makes a claim about how the human mind works, but has no reference to back it up.
- Your references lack a space between the parenthesis and the word prior, e.g., line 74: "input(Sundararajan et al., 2017)." -> "input (Sundararajan et al., 2017)."
- Have you explored the potential computational benefits of identifying non SK tokens and replacing them with mean replacement? You do have to compute gradients however, so I am unsure if there is a potential benefit here or not.

---

> ### Author Response · Authors · 2025-11-24
>
> We thank you for your constructive feedback. We address your questions as follows.
>
> **1. On contextualizing claims and generalizability**
>
> We have revised the manuscript to better contextualize our claims and to provide additional empirical evidence. In the Abstract and Experimental Setup (Sec. 4.1), we now explicitly describe BoolQ as our primary testbed, selected for its suitability for clean causal intervention and evaluation
> We further add supplementary experiments on RTE (textual entailment) and CNN/DailyMail (long-document summarization), reported in Sec. 4.1, Sec. 4.2.2, Table~1, and Appendix Sec. I, which show that the two-segment SK pattern also appears on these datasets.
> In the Conclusion (Sec. 5), we refine the phrasing of our main statements so that they explicitly refer to the empirical settings and tasks we study, and highlight that the observed SK patterns are supported by BoolQ, RTE, and CNN/DailyMail.
>
> **2. Responses to specific questions**
>
> - **Typo.** We corrected the reported typos throughout the paper.
> - **Appendix references.** All generic "see Appendix" mentions have been replaced by explicit references to sections.
> - **Lines 29--30 syntax.** The sentence at the beginning of the Introduction has been rewritten for clarity so that it now directly states our motivation and contributions (Sec. 1, lines 28--29 in revision).
> - **Line 35--36 (Human mind claim).** We retained the basic intuition but now support it with citations from cognitive science and decision-making research (van Dijk & Kintsch, 1983; Gigerenzer & Gaissmaier, 2011; Shah & Oppenheimer, 2008). The updated sentence appears in the Introduction (Sec. 1, lines 34--36 in revision).
> - **On potential computational benefits.** Our current SK identification procedure relies on gradient computation and an additional backward pass, so we use SK purely as a tool for analyzing how model decisions concentrate on a sparse subset of tokens, and only briefly mention potential efficiency or compression applications as future work in the saliency-aligned distillation discussion (Sec. 5 and Appendix Sec. I).

---

### Official Review · Reviewer_GkgX · 2025-11-03

**Soundness:** 3
**Presentation:** 3
**Contribution:** 3
**Rating:** 6
**Confidence:** 4

**Summary:**

This manuscript builds on the theoretical findings of this February 2024 paper: Tarzanagh et al. (2024) Transformers as Support Vector Machines, https://arxiv.org/abs/2308.16898. (Side note: There are two citations to the same paper in the References.)

The manuscript defines the "Support Knowledge" (SK) score for each token as the sum of (1) the gradient norm of the token's input embedding and (2) its attention-weighted max-margin score. Each component of the sum is weighted by a hyperparameter.

The manuscript claims that SK is a byproduct of the gradient-based optimization similar to how SVMs identify support vectors.

The manuscript contains experiments that show the importance of SK in binary QA classification.

Side note: There is a typo in Line 294.

**Strengths:**

- The topic of relating the attention mechanism to SVMs is interesting.
- The paper is easy to read.

**Weaknesses:**

- No theoretical explanation is provided for the two-segment inference mechanism observed in the experiments.

- This statement is a bit misleading: "This correspondence arises because backpropagation through transformer layers effectively solves the dual problem of an analogous SVM optimization." (Line 187-188)

After training, the optimal transformer attention layer can be mathematically represented as the dual expansion of an analogous SVM optimization. However, backpropagation operates in the primal parameter space, and the dual solution is a property of the model at optimum, not the direct result of the training algorithm

- A short summary of Appendix B should be added after Equation 5 to give the reader an idea of the importance of the values of alpha and beta.

- The explanation for why the values for alpha and beta in Equation 5 differ so much across various models (Lines 699-701) needs further investigation.

- There are a couple of missing related works:
(a) Tan M. Nguyen et al. A Primal-Dual Framework for Transformers and Neural Networks (https://arxiv.org/abs/2406.13781; June 2024)
(b) Shahar Katz and Lior Wolf. Reversed Attention: On The Gradient Descent Of Attention Layers In GPT (https://arxiv.org/abs/2412.17019; December 2024)

**Questions:**

- Can you provide a theoretical explanation for the two-segment inference mechanism seen in the experiments?

- Can you provide a more detailed explanation for why the values for alpha and beta in Equation 5 differ so much across various models?

---

> ### Author Response · Authors · 2025-11-24
>
> We thank you for your constructive feedback.
>
> **1. On the Quantitative Analysis of the Two-Segment Mechanism:**
>
> - **BoolQ stability statistics.** In the revised manuscript, we make the two‑segment inference mechanism explicit by reporting quantitative SK stability measures in Sec. 4.2.1. For Qwen3‑8B on a BoolQ sample, the top‑k Jaccard similarity of SK token sets between adjacent layers is low and unstable in early layers (layers 1–6, mean J ≈ 0.49, range 0.07–0.69), indicating an evidence‑gathering regime where the model keeps shifting attention across context tokens. Starting from layer 7, the Jaccard similarity becomes consistently high (layers 7–35, mean J ≈ 0.93, with many overlaps equal to 1.0), showing that the model has locked onto a nearly fixed SK set dominated by instruction and question tokens. Throughout this process, the effective SK token ratio remains very small (≈4–6% in layers 1–6 and ≈2.4% in layers 7–35).
>
> - **Cross‑task robustness.** In Sec. 4.2.2 and Appendix Sec in revision. We further report aggregate adjacent‑layer Jaccard statistics on RTE and CNN/DailyMail dataset. On RTE, the mean overlap rises from 0.62 in early layers (≈1–7) to 0.97 from layer 8 onward; on CNN/DailyMail, overlaps are ≈0.69–0.77 in early layers but close to 1.0 (mostly ≥0.97) in later layers (≈8–36). These quantitative results demonstrate that the same two‑segment SK pattern holds consistently across tasks.
>
> **2. On the Clarification of the SVM Analogy**
>
> We use the SVM framework as a conceptual reference. In the revised manuscript, Sec. 3.2 now emphasizes that the connection to SVMs is restricted to the shared linear scoring structure: attention weights and gradients are interpreted as dual‑like coefficients for identifying influential tokens, without stating that backpropagation solves an SVM dual problem. Moreover, Eq. (7) is explicitly presented as a first‑order Taylor approximation that yields a local sensitivity relation between gradients and the distance to an effective decision boundary, clarifying that our use of SVM terminology is an intuitive, local analogy rather than a global theorem.
>
> **3. On the Explanation for Varying α and β Values:**
>
> Our SK Score provides a unified framework by combining two components: causal sensitivity (measured by the gradient, α) and learned representational geometry (measured by the attention margin, β). The values of (α, β) represent the necessary parameterization of this framework for a given LLM's architecture, where a specific set of parameters should be determined to ensure optimal performance on each model.
>
> *   For instance, ChatGLM4 and Llama3.1 require a higher α. This suggests their architectures, such as the GLMBlock in ChatGLM, result in sharper decision boundaries where causal sensitivity is the dominant signal for identifying critical tokens.
> *   Conversely, the Qwen family requires a higher β. This indicates its design fosters more stable geometric representations, making the attention-based margin a more reliable indicator of importance.
>
> Therefore, the framework's validity is demonstrated by its adaptability to these architectural differences, not by a requirement for uniform parameters across all models.
>
> **4. On Other Revisions:**
> We will carefully revise based on your suggestions into the revision, including the references to Nguyen et al. (2024) and Katz and Wolf (2024), the summary of Appendix B, and the corrections for all typos and formatting errors.

---

### Meta-Review · Area_Chair_81NC · 2026-01-20

**Summary:**

Reviewers generally found the paper clearly written and the empirical results intriguing, but the concerns that drive my weak reject recommendation are about demonstrated impact. Two reviewers scored it slightly above threshold but were not strongly committed, and their feedback focused mostly on clarity and wording. In contrast, the rejecting reviewers raised more structural issues: the theoretical SVM framing is seen as under-defined and sometimes misleading (missing clear definitions/proof and ambiguous attention notation), the evaluation relies on relatively weak baselines and a single intervention setup that can feel bespoke, and also most importantly, at least for one reviewer, the work does not convincingly demonstrate downstream utility beyond diagnostic token importance (with only limited evidence that SK improves training). Taken together, these unresolved concerns outweigh the paper’s promising empirical observations in its current form. I believe a deeper iteration on the paper would be appreciated for a full acceptance.

**Reviewer Concerns:**

The rebuttal addresses several key points: it softens and clarifies the strongest SVM-related claims (responding to concerns about overstating the “dual” connection), adds quantitative evidence for the two-segment SK pattern, broadens the empirical scope beyond BoolQ with additional results on RTE and CNN/DailyMail, and provides an initial downstream demonstration via SK-weighted fine-tuning. However, important concerns remain: the theoretical framing is still not fully rigorous (unclear or inconsistent definitions/notation, specially around attention weights and “decision boundary” language and limited proof-level support; I believe with a full review process this could be addressed), the experimental validation still relies mainly on relatively weak baselines and a single intervention protocol, the $\alpha, \beta$ calibration varies widely across models without a strongly grounded explanation and may appear tuned to the evaluation, and the practical utility evidence is still narrow and modest relative to what the most critical reviewers expected (e.g., broader optimization benefits or stronger applications).

**Reviewer Scores:**

If they had participated fully in the discussion, I expect modest upward shifts for most reviewers: GkgX would likely move from 6 to 7 and KSdf from 6 to 7 given the clarifications and added experiments, Wss1 might increase from 4 to 5 due to improved framing and additional evidence, while NE1S would likely remain skeptical but could move from 2 to 3 after the added SK-weighted fine-tuning result.
This is my most optimistic guess after deep discussions and revisions of the paper.

---

### Decision · Program_Chairs · 2026-01-26

Reject